# A Novel Device for Blood Drainage after Le Fort I Osteotomy: Maxillary Sinus Ventilation Drainage (MSVD)

**DOI:** 10.3390/jcm11030562

**Published:** 2022-01-23

**Authors:** Ui-Lyong Lee, Hyo-Won Jang, Han-Wool Choung, Sei-Young Lee, Young-Jun Choi

**Affiliations:** 1Department of Oral and Maxillofacial Surgery, Dental Center, Chung-Ang Hospital, Seoul 06973, Korea; davidjoy76@gmail.com (U.-L.L.); woolmania@naver.com (H.-W.C.); 2Department of Oral and Maxillofacial Surgery, Yonsei Twins Dental Clinic, Seoul 07997, Korea; ystwin1@naver.com; 3Department of Otorhinolaryngology-Head and Neck Surgery, College of Medicine, Chung-Ang University, Seoul 06973, Korea; syleemd@cau.ac.kr; 4Department of Oral and Maxillofacial Surgery, College of Medicine, Chung-Ang University, Seoul 06973, Korea

**Keywords:** maxilla, osteotomy, drainage, hematoma, maxillary sinus, ventilation

## Abstract

The purpose of this study is to present a novel maxillary sinus ventilation drainage (MSVD) device which facilitates blood drainage and nasal breathing after Le Fort I osteotomy. One hundred patients who underwent bimaxillary orthognathic surgery from January 2016 to June 2016 at the Department of Oral and Maxillofacial Surgery, Chung-Ang University Hospital were retrospectively selected and divided into two groups. MSVD was applied in 50 patients, who were allocated to the MSVD group, while the remaining 50 patients, in whom MSVD was not applied, were allocated to the non-MSVD group. All patients underwent a cone-beam computed tomography (CBCT) scan before and 2 days after surgery. CBCT was used to analyze middle meatus patency and the percentage of hematoma volume per entire maxillary sinus volume. Statistical comparisons between the two groups were performed using the Chi-squared and Mann–Whitney U tests to investigate the clinical effectiveness of MSVD. The MSVD group showed significantly higher maintenance ratio of the middle meatus patency and a higher percentage of maxillary sinus air volume (*p* < 0.05) than the non-MSVD group. MSVD facilitated nasal breathing after Le Fort I osteotomy by reducing hematoma inside the maxillary sinus and promoting middle meatal patency.

## 1. Introduction

Since von Langenbeck first described Le Fort I osteotomy in 1859, it has become a routine procedure for bimaxillary orthognathic surgery [1,2]. Previous studies demonstrated that this osteotomy is generally a safe and predictable procedure [3,4]. Nevertheless, postoperative complications related to hemorrhage are prevalent [3].

Postoperative hematoma is a suitable environment for the growth of bacteria and increases the risk of postoperative infection [5]. Drainage of the hematoma in a surgical wound is considered effective in reducing infection [6]. Nasal stuffiness is another hemorrhagic complication after Le Fort I osteotomy caused by sino-nasal hematoma, which blocks the middle meatus. According to questionnaires regarding postoperative patient complaints, the worst subjective symptom after bimaxillary orthognathic surgery is nasal stuffiness, with a frequency of about 55.6% [7]. Furthermore, postoperative oozing from the maxillary sinus might favor the formation of endobronchial blood clots through the nasopharynx, which may result in airway obstruction [8]. Therefore, appropriate blood drainage from the maxillary sinuses is necessary to reduce such complications.

However, the maxillary sinus is not a completely enclosed space. This chamber communicates with the nasal cavity through an ostium that typically opens into the middle nasal meatus [9]. Therefore, the use of vacuum suction drainage, for example a Hemovac drain (Zimmer, Warsaw, USA), cannot maintain the interior of the maxillary sinus under negative pressure.

Hence, Professor Young-Jun Choi invented the novel maxillary sinus ventilation drainage (MSVD) device, which facilitates blood drainage, thereby reducing hematoma and helping to quickly regain nasal breathing after Le Fort I osteotomy. In this article, we would like to present the procedure and effects of MSVD after bimaxillary orthognathic surgery.

## 2. Materials and Methods

### 2.1. Study Designs

This study evaluated 100 patients who underwent bimaxillary orthognathic surgery from January 2016 to June 2016 at the Department of Oral and Maxillofacial Surgery, Chung-Ang University Hospital. The patients were retrospectively divided into 2 groups; 50 patients were allocated to the non-MSVD group, and the other 50 patients to the MSVD group. One oral and maxillofacial surgeon performed all bimaxillary orthognathic surgeries. This study was approved by the Clinical Research Committee of Chung-Ang University College of Medicine (IRB #1907-005-16271).

To be included in the study sample, patients were assessed through cephalometric analysis, and received orthodontic treatment before and after bimaxillary orthognathic surgery, which was composed of Le Fort I osteotomy and short-lingual osteotomy of the mandible. During the maxillary surgery, all descending palatine arteries were secured, and the Schneiderian membrane was managed as conservatively as possible. Thereafter, the maxilla was fixed with mini-plates (two 4-hole mini-plates, four 3-hole mini-plates) and screws. Postoperatively, antibiotics (amoxicillin sodium with clavulanate potassium), methylprednisolone, and NSAIDs (ketorolac tromethamine) were administered intravenously at the proper dosage. The operation time was no more than 3 h, and ages of patients ranged from 20 to 30 years old. 

Patients were excluded as study subjects if they met any of the following exclusion criteria: medically compromised patients (for example hypertension, diabetes mellitus, coagulation problems), patients with history of maxillary sinus surgery, cases of complicated maxillary surgery like segmental osteotomy, and patients with bleeding or coagulation disorders. None of the patients included in this study were under treatment with significant anticoagulant or antiplatelet medicines.

The MSVD device is composed of two drainage tubes for each maxillary sinus. The “IN tube” for air inflow or irrigation is a 10 Fr silicone suction catheter, and the “OUT tube” for suction drainage is 8 Fr (Figure 1). After maxillary fixation and alar cinch suture, 2 round holes (each with a diameter of about 4 mm, the same as the MSVD tube diameter) were drilled on each side of the anterior maxilla above the osteotomy line to place the IN and OUT tubes (Figure 2a–c). Through these holes, both the IN and OUT tubes were placed in the middle of the maxillary sinus. The mean distance from the upper posterior vestibule to the center of maxillary sinus has been reported to be approximately 7 cm in South Koreans [10]. A mark with surgical tape (Ioban^®^, 3M Medical, St. Paul, MN, USA) was placed 7 cm from the inner ends of the tubes, so that the inner ends of the tubes were located at the center of the maxillary sinus (Figure 2d,e). The outer parts of all tubes were positioned at the posterior mucosal suture lines and initially fixed under tight continuous sutures (Figure 2f). After suturing of the intraoral mucosal wounds, the site marked with the surgical tape on the tubes was firmly tagged on the vestibular mucosa with suture material. 

The outer ends of the two OUT tubes were gathered, inserted into a round plastic lid (plastic connecting portion of the silicon suction catheter), and fixed with Ioban tape, so that 1 wall suction could be connected to both OUT tubes via the single plastic lid (Figure 3a, box). Under a continuous negative pressure of 100 to 150 mmHg after connecting to wall suction, the blood accumulated in the maxillary sinus could be easily discharged (Figure 3b, box). On the other hand, intravascular catheters were inserted into the outer ends of both IN tubes to facilitate irrigation with syringes postoperatively, and fixed with Ioban tape (Figure 3a, arrow). Next, the outer ends of the IN tubes were loosely covered with 2 × 2 gauze (except during irrigation), which served as an air filter (Figure 3b, arrow). If the gauze gets wet with blood or saliva, it must be replaced; otherwise, external air cannot be introduced and the negative pressure in the maxillary sinus may become excessively high. 

To prevent infection in the maxillary sinus and blockage of the IN and OUT tubes with blood clot, each maxillary sinus should be irrigated with a mixture of 100 mL of normal saline and 50 mL of 0.12% chlorhexidine gluconate solution every 2 h after surgery. All tubes should be removed the day after operation. However, if blood drainage through the OUT tube does not decrease, it is better to maintain the tubes for another day. 

### 2.2. Radiological Analysis

All patients underwent cone-beam computerized tomogram (CBCT, Kavo 3D exam, 37.10 mAs, 120 KVP, acquisition time 17.8 s) before and 2 days after bimaxillary orthognathic surgery. One of the authors analyzed the radiological features using 3D viewer software (Invivo^®^, Anatomage, San Jose, CA, USA). The middle meatus patency and the ratio of air volume per total maxillary sinus volume were also determined. On coronal view, the patency from the maxillary sinus to the nasal airway was evaluated (Figure 4). The entire maxillary sinus volume before the surgery was analyzed by CBCT using a 3D-volume measuring system for the right and left sides separately. The air volume of the maxillary sinus was evaluated 2 days after surgery with the same method. Further, air volume as per the entire maxillary sinus volume was calculated (Figure 5).

### 2.3. Statistical Analysis

This study included 100 patients who received routine bimaxillary orthognathic surgery from January 2016 to June 2016. Levene’s test was performed to evaluate the difference between the mean value and variance in both groups. Both groups were analyzed with the Chi-squared and Mann–Whitney U tests to determine the presence of statistically significant differences in middle meatus patency and air volume/total volume ratio. Statistical analyses were performed using SAS version 22 (SAS Institute Inc., Cary, NC, USA), and the results were considered statistically significant when *p* < 0.05.

## 3. Results

All patients in this study underwent bimaxillary orthognathic surgery. The MSVD group consisted of 50 subjects (22 men and 28 women) with a mean age of 25.98 years (men 25.27 years; women 26.54 years). The non-MSVD group consisted of 50 subjects (27 men and 23 women) with a mean age of 27.42 years (men 27.93 years; women 26.83 years). There were no significant differences between the two groups in mean value and variance for sex, age, operation time, estimated blood loss, and amount of maxillary movement (Table 1 and Table 2). Regarding middle meatus patency and air/total volume ratio of the maxillary sinus, the Chi-squared test showed a more open tendency of the middle meatus in the MSVD group (58%) than in the non-MSVD group (29%) (*p* < 0.001, Table 3). The Mann–Whitney U-test showed that the air/total volume ratio of the maxillary sinus was considerably higher in the MSVD group than in the non-MSVD group (*p* < 0.001, Table 4).

## 4. Discussion

One of the most common complaints of patients after Le Fort I osteotomy is nasal stuffiness [7]. To help the patient breathe through the nose, special equipment such as nasotracheal or nasopharyngeal airway tubes are applied. However, these are usually irritating and unbearable when patients are awake. The MSVD facilitates nasal breathing more effectively than either nasotracheal or nasopharyngeal airway tubes [11]. The MSVD removes the hematoma in the maxillary sinus and opens the middle meatus, allowing comfortable nasal breathing and preventing epistaxis.

Furthermore, maxillary bleeding can be easily monitored through the MSVD. This allows surgeons to determine whether additional bleeding control is required. On the first day after surgery, when blood drainage through the MSVD has almost stopped, all tubes must be removed. However, if bleeding is still observed through MSVD, tranexamic acid or fresh-frozen plasma should be administered. If a large amount of blood comes out through the tubes, additional hemostasis should be considered to reduce severe hemorrhagic complications [12].

Regarding the changes in pan-sinus dynamics under the MSVD system on postoperative day 1, no additional bleeding in the maxillary sinus occurred due to MSVD, and no specific side effects related to sinus functions were observed after the surgery. According to our preliminary data, some patients showed mild ear or nose discomfort at negative pressures under 150 mmHg. Thus, in the present study, a continuous negative pressure of 100 to 150 mmHg was applied, and the result showed that this condition is enough to drain the sinus blood without obstruction of the tubes and without causing patient discomfort related to the MSVD.

Retrograde infection is a problem linked to the use of devices for wound drainage. However, the optimum period for retention of drains is still controversial [13]. In this study, use of MSVD for 1 day (maximum 2 days) with irrigation of chlorhexidine gluconate solution every 2 h after surgery did not lead to retrograde infection. Sino-nasal irrigation with normal saline is a simple treatment that relieves symptoms of a variety of sino-nasal conditions and has been advocated for postoperative cleaning of the nasal cavity [14]. Chlorhexidine gluconate is a broad-spectrum antiseptic agent that has been proven to be effective against gram-positive and gram-negative bacteria, as well as selective fungi [15]. Since the Food and Drug Administration of the United States approved the use of 0.05% chlorhexidine gluconate for irrigating and cleaning surgical wounds in 2012, wound irrigation with aqueous chlorhexidine gluconate has shown therapeutic benefits because it is not inactivated by blood or tissue protein [16]. We used 0.04% chlorhexidine gluconate for sinus irrigation every 2 h after the surgery, with favorable results.

During the Le Fort I osteotomy, autogenous block bones were grafted at the bony gap of the anterior maxilla for postoperative stability of the maxilla and for achieving a more efficient drainage of the maxillary sinus by minimizing the communicating spaces (Figure 6).

Questions may arise as to whether the bony holes drilled to apply the MSVD will heal completely with time. We observed that drilled bony holes had become smaller during the metal removal surgery at 2 years after bimaxillary orthognathic surgery (Figure 7). These remaining bony defects did not clinically affect the physiology or function of the maxillary sinus. At the last follow up, 4 years after the procedure, there were no other sino-nasal clinical symptoms or complications regarding the MSVD. All tubes should be removed the day after operation to prevent the active post-operative bleeding. According to our experience, there was no case of late bleeding in the group of patients with MSVD. However, it is important to alertly monitor for late bleeding.

Conventional surgical drains passively drain blood, but MSVD actively suctions blood in the maxillary sinus, preventing blood from accumulating in the maxillary sinus and thus preventing nasal blockage due to backflow of blood into the nasal cavity.

In our experience, contra-indications for MSVD include when the maxillary bleeding during surgery is very little, a segmental oseotomy is performed, and when the maxilla is moved downwards and a bone graft is applied to the bone gap.

At postoperative day 2, radiographic analysis by CBCT showed that the MSVD group had a higher maintenance ratio of middle meatus patency and a higher percentage of maxillary sinus air volume than the non-MSVD group. Therefore, this study suggests that the MSVD allows nasal breathing immediately after Le Fort I osteotomy. We expect that the MSVD will be applied in other maxillary sinus surgeries such as Caldwell-Luc operations.

Future studies should confirm the appropriate suction pressure, the most effective concentration of chlorhexidine solution, and the appropriate frequency of chlorhexidine irrigation. The limitation of this research is that the study was based on purely radiologic findings and not on clinical symptoms. However, studies have shown that the most common complaint of patients with hematoma of the maxillary sinus is epitaxis, followed by nasal obstruction and facial pain. Consequently, MSVD can facilitate nasal breathing after Le Fort I osteotomy by reducing hematoma inside the maxillary sinus [17,18].

## Figures and Tables

**Figure 1 jcm-11-00562-f001:**
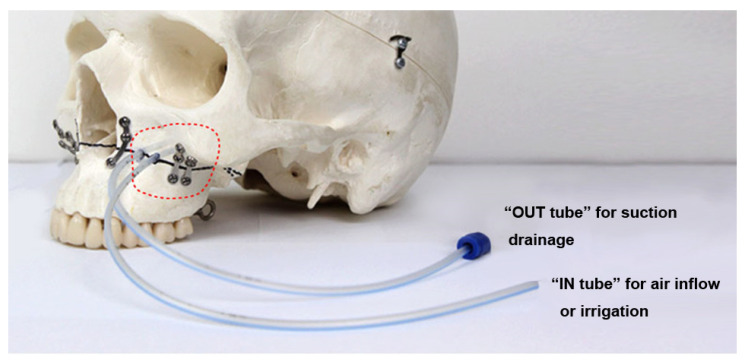
A schematic diagram of maxillary sinus ventilation drainage (MSVD). The MSVD device is composed of 2 drainage tubes for each maxillary sinus: the “IN tube” for air inflow or irrigation and the “OUT tube” for suction drainage. Round holes are drilled on the anterior wall of the maxilla, and the MSVD tubes are inserted into the maxillary sinus via these holes.

**Figure 2 jcm-11-00562-f002:**
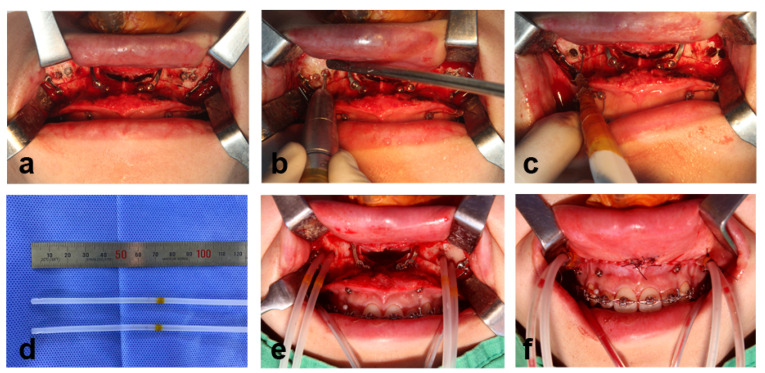
The surgical procedures for maxillary sinus ventilation drainage (MSVD). (**a**) Location of sinus holes for the MSVD. (**b**) Sinus holes were drilled using a round bur. (**c**) Fenestration of sinus membrane and hemostasis using a Bovie. (**d**) The surgical drape Ioban was wrapped around the tube at about 7 cm from the inner tube end, indicating the depth of tube insertion. (**e**) MSVD tubes were inserted into the maxillary sinuses. (**f**) The mark of the tubes was firmly tagged on the vestibular mucosa with suture material. Both the IN tube ends and OUT tube ends were attached to each other using Ioban tape.

**Figure 3 jcm-11-00562-f003:**
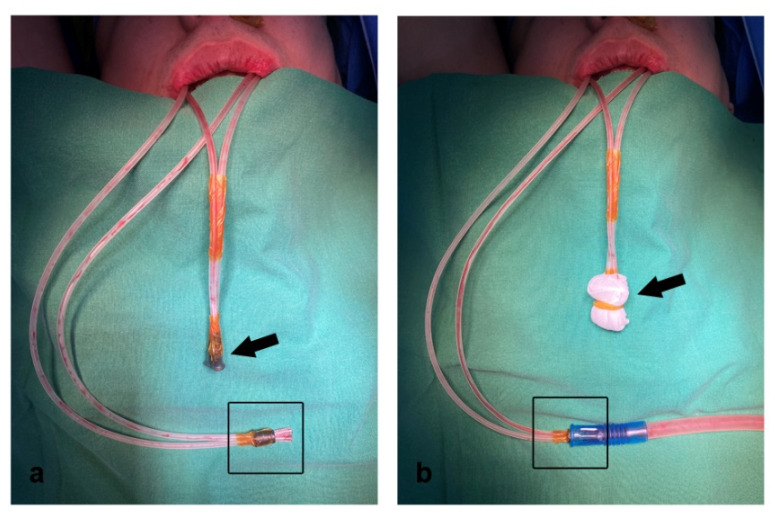
Outer ends of both the IN (arrows) and OUT tubes (box). (**a**) Intravascular catheters were inserted into the outer ends of both IN tubes and fixed with Ioban tape (arrow). The outer ends of both OUT tubes were gathered, inserted into a plastic lid, and fixed with Ioban tape (box). (**b**) The outer ends of the IN tubes were loosely covered with a 2 × 2 gauze, which served as an air filter (arrow). One wall suction can be connected to both OUT tubes via the single rubber lid (box).

**Figure 4 jcm-11-00562-f004:**
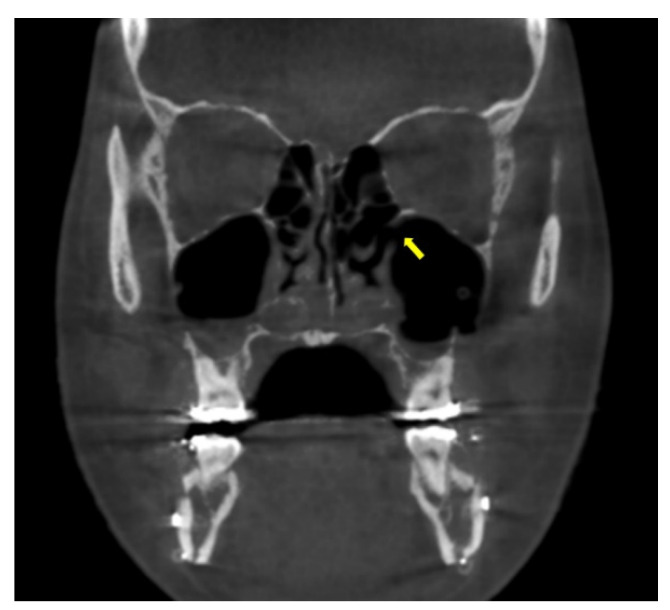
The patency of middle meatus was confirmed on coronal view of cone-beam computed tomography 2 days after surgery (arrow).

**Figure 5 jcm-11-00562-f005:**
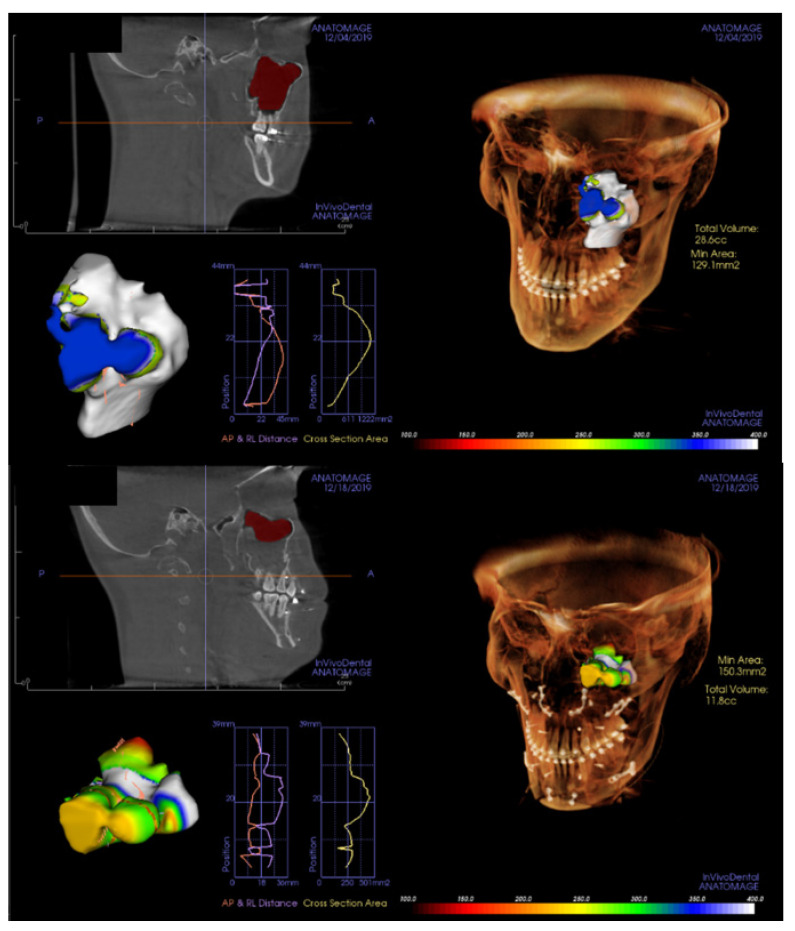
The entire maxillary sinus volume before the surgery was checked by cone-beam computerized tomogram, using a 3-dimensional volume measuring system (upper). The air volume of maxillary sinus 2 days after surgery was checked with the same method (lower). Then, the ratio of air volume per total maxillary sinus volume (%) was calculated for the left and right sides separately.

**Figure 6 jcm-11-00562-f006:**
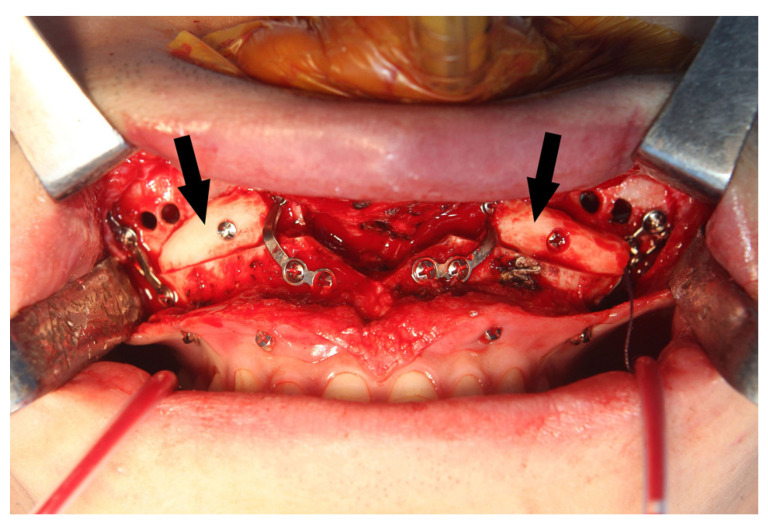
Autogenous block bone (arrows) was grafted at the bony gap of maxillary sinus wall for a more efficient drainage of maxillary sinus.

**Figure 7 jcm-11-00562-f007:**
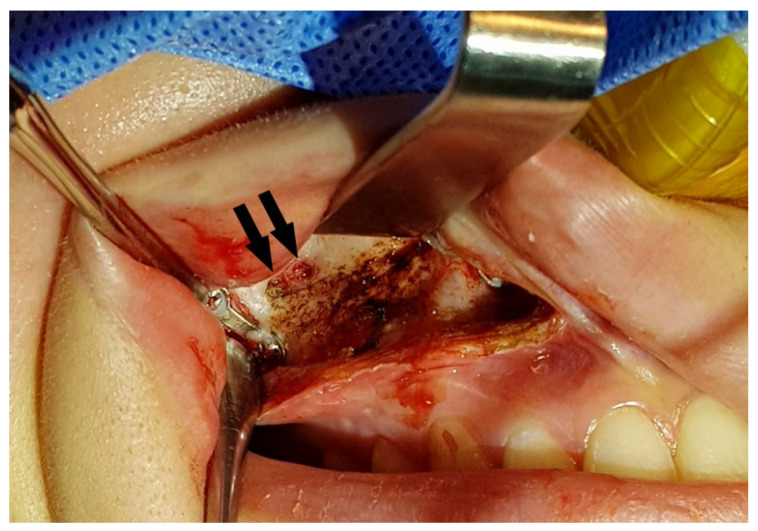
The drilled bony holes for the maxillary sinus ventilation drainage had become smaller (arrows) at 2 years after the bimaxillary orthognathic surgery.

**Table 1 jcm-11-00562-t001:** Subject allocation in the non-MSVD and MSVD groups.

	Sex	Age (year)	Post Imp (mm)	Mx Cant (mm)	Op Time (h)	EBL (cc)
**Non-MSVD group**	Male (*n* = 22)	25.27 ± 3.55	5.91 ± 3.41	2.01 ± 1.19	2.82 ± 0.47	864.29 ± 127.62
	Female (*n* = 28)	26.54 ± 5.6	7.26 ± 2.81	1.27 ± 0.75	2.76 ± 0.5	857.5 ± 117.29
	Total (*n* = 50)	25.98 ± 4.8	6.55 ± 3.17	1.53 ± 0.97	2.79 ± 0.48	860.98 ± 121.2
**MSVD group**	Male (*n* = 27)	27.93±4.02	5.35 ± 2.85	1.89 ± 1.13	2.66 ± 0.41	810.87 ± 207.78
	Female (*n* = 23)	26.83±3.65	6.61 ± 3.04	1.89 ± 2.29	2.57 ± 0.45	772.73 ± 177.77
	Total (*n* = 50)	27.42±3.85	6.01 ± 2.98	1.89 ± 1.67	2.62 ± 0.43	792.22 ± 192.46

EBL, estimated blood loss; MSVD, maxillary sinus ventilation drainage; Mx cant, maxillary canting; Op time, operation time; Post imp, posterior impaction

**Table 2 jcm-11-00562-t002:** Independent samples test in the non-MSVD and MSVD groups.

	Levene’s Test for Equality of Variances		t-Test for Equality of Means						
	F	Sig	t	df	Sig (2-Tailed)	Mean Difference	Std. Error Difference	95% Confidence Interval of the Difference	
								Lower	Upper
**Post imp**	0.138	0.711 *	1.17	100	0.245	0.811	0.693	−0.568	2.191
**(mm)**									
**Mx cant**	1.622	0.208 *	−1.103	100	0.275	−0.422	0.382	−1.187	0.344
**(mm)**									
**Op time**	0.334	0.565 *	1.756	100	0.083	0.172	0.098	−0.0228	0.367
**(h)**									
**EBL**	0.865	0.355 *	1.96	100	0.053	68.753	35.077	−1	138.509
**(cc)**									

EBL, estimated blood loss; MSVD, maxillary sinus ventilation drainage; Mx cant, maxillary canting; Op time, operation time; Post imp, posterior impaction; * *p* > 0.05.

**Table 3 jcm-11-00562-t003:** Chi-squared test for patency of middle meatus (non-MSVD group and MSVD group).

		**Middle Meatus**		**Total**	
		**Closed**	**Open**		
**Non-MSVD group**	Count	71	29	100	
	Expected Count	55.7	44.3	100.0	
	%	71.2	28.8	100.0	
**MSVD group**	Count	42	58	100	
	Expected Count	55.8	44.2	100	
	%	42.2	57.8	100.0	
**Total**	Count	113	87	200	
	Expected Count	111.0	89.0	200.0	
	%	55.8	44.2	100.0	
**Test Statistics**					
	**Value**	**df**	**Asym. Sig** **(2-sided)**	**Exact Sig** **(2-sided)**	**Exact Sig** **(1-sided)**
**Pearson Chi-square**	13.301	1	0.000		

MSVD, maxillary sinus ventilation drainage.

**Table 4 jcm-11-00562-t004:** Air/total volume ratio of maxillary sinus (non-MSVD group and MSVD group).

		** *n* **	**Mean Rank**	**Sum of Ranks**
**Air/total volume ratio**	**Non-MSVD group**	100	77.83	7783.00
	**MSVD group**	100	123.17	12317.00
	**Total**	200		
**Test Statistics**				
			**Air/total volume ratio**	
**Mann–Whitney U**			2733.000	
**Wilcoxon W**			7783.000	
**Z**			−5.539	
**Asym. Sig (2-sided)**			0.000	

MSVD, maxillary sinus ventilation drainage.

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
