# Peer review of "A Novel Device for Blood Drainage after Le Fort I Osteotomy: Maxillary Sinus Ventilation Drainage (MSVD)"

_jcm, 2022, doi:10.3390/jcm11030562_

Round 1

Reviewer 1 Report

  I would like to congratulate the authors for this study, but I would like to suggest some improvements.

  In the materials and methods section, authors should include whether or not the patients were using any anticoagulant/antiplatelet medication, such as Apixaban or Clopidogrel.

  For the analysis, I suggest that the authors include cases using conventional surgical drains or other procedure/medications to aid sinus drainage, and also include new post-surgical CBCT after 3-4 weeks to have better data to support their findings.

  For the discussion section, the authors need to better explain and justify the results of this study.

  I suggest the authors discuss the possibility of late bleeding episodes and also in this section it is important to discuss the differences, benefits and contraindications of using MSVD compared to others drainage devices, procedures or medications.

Reviewer 2 Report

The authors have performed a retrospective study comparing the outcome of 50 patients treated with maxillary sinus ventilation drainage (MSVD) with 50 patients treated with conventional orthognathic surgery.

The study design seems to be comparative cohorts included retrospectively.

The outcome was air in the maxillary sinus evaluated on postoperative CBCT scans.

There is a problem with the study design, that requires additional justification before the manuscript is accepted for publication.

The clinical impact of this study is not determined. In the introduction, the authors argue point to the patients perceived nasal stuffiness and postoperative oozing to be the worst subjective symptoms following bimaxillary surgery. However, the study is purely radiological and as such, it does not measure if the patients actually perceived any benefits from the MSVD intervention. This is really the largest limitation in this study, that the study is purely radiological without the patient’s perspective that the intervention is aimed to improve.

A few minor areas for revision:

The patency of the middle meatus was evaluated using CBCT, but the authors do not describe how it was judged if the sinus was filled with blood. Since the patients’ experience oozing from the maxillary sinus, is the meatus really blocked?

In the discussion, the authors suddenly claim that the patients in the MSVD-group had less swelling in the midface, but in the results section, this is not evident. Either the data should be shared with the readers, or this new results/discussion should be left out.

Round 2

Reviewer 1 Report

Once again I would like to congratulate the authors for this study, but unfortunately the paper continue to need some adjustments before publish.

Due the possibility to have late bleeding and to better analyze the effectiveness of the MSVD I continue to suggest the authors to include a CBCT  3-4 weeks post surgical procedure, including with new statistical analysis.

At discussion section the authors need to improve the scientific basis and based on that improve the discussion compared with others devices, procedures or medications. It is also important to include here the possibility to increase/promote active post-operative bleeding, specially in cases with pseudoaneurysm formation in the maxillary artery.

Reviewer 2 Report

The authors have performed a substantial revision and the manuscript has been improved substantially.

Author Response

Thank you so much.